# Design for Distributed Feedback Laser Biosensors Based on the Active Grating Model

**DOI:** 10.3390/s19112569

**Published:** 2019-06-05

**Authors:** Bowen Wang, Yi Zhou, Zhihe Guo, Xiang Wu

**Affiliations:** Key Laboratory of Micro and Nano Photonic Structures (Ministry of Education), Department of Optical Science and Engineering, Shanghai Engineering Research Center of Ultra Precision Optical Manufacturing, Fudan University, Shanghai 200433, China; 16210720012@fudan.edu.cn (B.W.); 18110720008@fudan.edu.cn (Y.Z.); 17110720004@fudan.edu.cn (Z.G.)

**Keywords:** distributed feedback laser, biosensor, active grating

## Abstract

The distributed feedback (DFB) laser is widely used in sensing because of its portable size, simple fabrication and high sensitivity. Most theoretical design models are based on passive Bragg gratings. However, passive grating models cannot be used to predict sensor performance using the important indicator of figure of merit (FOM) through theoretical calculations. To solve this problem, we replaced the passive grating with an active grating by using the imaginary part of the coupling constant that represents the value of the gain. As a comparison, the influence of the full width at half maximum (FWHM) and sensitivity were analyzed for different grating duty cycles and depths in the passive grating sensors. To obtain a higher FOM in the active grating sensors, we systematically investigated the effects of duty cycle and gain value through numerical simulations. We found that the redshift caused by a duty cycle increase can improve the sensitivity of biomolecule detection by 1.7 times.

## 1. Introduction

Optical biosensors based on the distributed feedback (DFB) laser, with their advantages of small and portable size, simple fabrication, low cost and high sensitivity, have attracted increasing attention and represent one of the most important tools for drug screening, diagnostic testing, nucleic acid detecting, food safety and environmental monitoring [1,2,3,4]. DFB lasers obtain feedback via backward Bragg scattering from periodic perturbations of the refractive index and/or gain [5]. For a DFB laser sensor, the resonant wavelength is described by the Bragg equation [6]:(1)$$m\lambda =2\Lambda n_{eff}$$ where *m* is the order of diffraction, *λ* is the wavelength of the laser, *n*_eff_ is the mode’s effective refractive index, which for a specific mode is a function of the refractive index of the surrounding medium [7], and *Λ* is the grating period. When the order of diffraction is 2 (*m* = 2), the laser will emit vertically from the surface [8]. Due to the dependency of *n*_eff_ on the surrounding medium, any adsorption and binding of bio or chemical analyte to the sensor changes the effective refractive index, causing a shift in the wavelength of the resonant mode. This change in the resonant condition can be determined using a spectrograph. The shift in wavelength is detected as a signal of linkage between the surface and analyte; thus, the wavelength sensitivity can be obtained and the mass of bound molecules can be calculated [9,10].

To optimize DFB sensors, numerous methods have been investigated. Researchers have replaced the substrate with a low-index porous dielectric [11] or UV-curable polymer [12] and included a high refractive index thin film such as titanium dioxide [10,13]. The electric field will ascend from the substrate to the grating and the surrounding medium, as a result of reducing the refractive index of the substrate and incorporating high-index thin films. DFB sensors are more sensitive to changes to the refractive index of the cover medium, thereby achieving the purpose of improving the sensor’s sensitivity. To increase the reaction area, Ge et al. replaced the high-refractive-index thin film with high-refractive-index nanorod structures on the surface of the sensor [14]. The larger reaction area can link more antibodies that can specifically recognize the antigen in biosensing, which can reduce the detection limit of the antigen. Others have used different active films, including dye-doped polymers [15,16,17,18] and organic semiconductors [19,20,21,22,23,24,25], for bulk and bio-detection. 

In sensing, sensitivity (S) is defined as the amount of shift in the resonant wavelength (Δ*λ*) caused by a change in the refractive index (Δ*n*) or thickness (Δ*d*) of the cover medium (S = Δ*λ*/Δ*n* or S = Δ*λ*/Δ*d*). The figure of merit (FOM) is used to evaluate the performance of label-free biosensing [26]. This indicator is defined as the ratio of the sensitivity (S) of the sensor and the full width at half maximum (FWHM) of the resonant wavelength (FOM = S/FWHM) [27]. To simulate the performance of a sensor, an effective design model is necessary before a sensor is fabricated. However, most theoretical design models are based on passive Bragg gratings. Thus, the FWHM of the resonant peak calculated by the passive model must be larger than the laser sensors, and passive grating models cannot evaluate sensor performance using the important indicator of the FOM. In addition, many works only focused on changing the analyte’s influence on the effective refractive index. Although the effective refractive index could influence the resonant wavelength, the grating parameters are also important. Therefore, an active grating model of DFB laser sensors, which could predict the FOM and consider the influence of grating parameters, is necessary. 

In this work, we first investigated the influence of the FWHM, sensitivity and FOM on passive grating sensors by changing the grating groove depth (*d*_g_) and duty cycle (DC) in numerical simulations. To more accurately simulate the performance of DFB laser sensors, we replaced the passive grating with an active grating and obtained the relationships between the gain value, relative intensity and FWHM. Moreover, we found that gratings with a DC of more than 50% can improve the sensitivity of biomolecule detection in sensors. Finally, the variation of the laser intensity with the thickness of the biomolecule layer (*d*_bl_) was observed. 

## 2. Simulation Results and Discussion

The RSoft product based on rigorous coupled-wave analysis (RCWA) [28] was used to simulate the effect of the grating DC and *d*_g_ on the sensors’ sensitivity, FWHM and FOM. COMSOL software, which is based on the finite element analysis (FEA) method [29], was used to calculate the electric field distributions. In the following work, we simulated passive grating sensors and active grating biosensors made of a high index layer (*d*_hil_ = 30 nm, *n*_hil_ = 2.50), with ultralow refractive index porous SiO_2_ (*n_sub_* = 1.09) [30], which was immersed in a liquid medium (*n_c_*), as the substrate. *d*_hil_ and *d*_sub_ are the thickness of high index layer and substrate, respectively. The substrate had a set thickness of 2400 nm and was regarded as infinite compared to the other layers (*d*_sub_ = 2400 nm). The refractive indexes of both the grating and waveguide layers were 1.59 (*n*_g_ = *n*_wg_ = 1.59). The grating period (*Λ*) was 400 nm.

### 2.1. The Effect of DC and d_g_ on Passive Grating Sensors

The effects of DC and *d_g_* were thoroughly investigated to analyze the performance of the passive grating sensors. The DC of a grating is defined as the length of the ridge (*d*_r_) divided by the period (*Λ*) of the grating (DC = *d*_r_/*Λ*). A schematic of the passive grating sensors is depicted in Figure 1a. The total thickness of the grating and waveguide layers is 200 nm. *d*_wg_ is the thickness of the waveguide layer. *d_g_* was chosen as 50, 100, 150 and 200 nm with corresponding *d*_wg_ values of 150, 100, 50 and 0 nm. The DCs of the grating were set to 20, 30, 40, 50, 60, 70 and 80%. The green and blue arrows represent the first-order and second-order diffractive direction of the light in the structure, respectively. The first-order and second-order diffracted light was detected from the waveguide. In the vertical direction from the surface, we could detect the second-order diffracted light.

When the cover medium is water (*n*_c_ = 1.33), the FWHM of the resonant peak is as illustrated in Figure 1b. As *d*_g_ increases, the FWHM of the resonant peak gradually becomes wide for all DCs except 70 and 80%. The reason for this result is that a deeper *d*_g_ could diffract more energy of the resonant mode out of the structure and thus lead to a larger FWHM. However, when the DC is 70 and 80%, the gap between the ridges is too narrow to diffract light. For every *d*_g_, as the DC increases, the FWHM first widens and then becomes narrow when DC = 50%. The narrowest FWHM of 2.22 nm is achieved when *d_g_* is 50 nm (*d*_wg_ is 150 nm) and the DC is 20%. As observed in Figure 1c, the resonant wavelength (*λ*_peak_) shifts when the *n*_c_ is changed. Through fitting *λ*_peak_ and *n*_c_, which is shown in Figure 1c, we obtain the bulk sensitivity as shown in Figure 1d.

Figure 2a demonstrates the relationship between DC, *d*_g_ and sensitivity. For the same *d*_g_, the sensitivity change is small for varying DCs. In contrast, it is obvious that the sensitivity increases rapidly with the deepening of *d*_g_. The sensitivity of the deepest grating (*d*_g_ = 200 nm) is about twice that of the shallowest (*d*_g_ = 50 nm). When *d*_g_ is increased (*d*_wg_ is decreased), the liquid under the test can go deep into the structure, leading to a stronger reaction to light. As shown in the electric field distributions of the structures with a *d*_g_ of 50 and 200 nm in Figure 2b, more energy leaks into the cover medium when *d*_g_ = 200 nm. At the same time, the maximum electric intensity (|E|_max_) obtained when *d*_g_ = 200 nm (2.02 × 10^5^ V/m) is lower than that obtained when *d*_g_ = 50 nm (3.52 × 10^5^ V/m). Therefore, the sensors would become more sensitive with a refractive index change in the cover medium. The three maximum sensitivities of 218.2, 217.3 and 211.2 nm/RIU are achieved when *d*_g_ is 200 nm (*d*_wg_ = 0 nm) and the DCs are 40, 50 and 60%, respectively. The more energy that leaks on to the liquid would broaden the FWHM. In Figure 2c a schematic diagram is shown to make the waveguide structure more clear. In Figure 2d, the FOM decreases as *d_g_* increases. For the same *d*_g_, the variation tendency of the FOM with DC is completely opposite to the trend of the FWHM with DC. This means that the FWHM has a greater impact on the value of the FOM. The maximum FOM of 45.13 RIU^−1^ is obtained when *d*_g_ is 50 nm (*d*_wg_ = 150 nm) and DC is 20%. Figure 2e illustrates the relationship between the resonant wavelength and grating DC when *d_g_* was set to 200 nm. As observed, the wavelength is almost unchanged with increasing DC below 50%. Once the value of the DC exceeds 50%, the wavelength sharply shifts in the long wavelength direction. 

### 2.2. Active Grating for High FOMs of Sensors

The FOM is a good indicator for measuring the ability of a sensor [26]. This value is closely related to the FWHM and sensitivity of the sensor. To enhance the value of the FOM, we could approach from two directions: decreasing the FWHM and/or increasing the sensitivity. Higher sensitivity requires a more intense interaction between light and the surroundings, which means more light energy needs to leak into the cover medium so that the FWHM will broaden. A large FWHM limits the capability of a sensor to precisely measure small resonant wavelength shifts [27]. However, a narrow FWHM requires light energy that is mostly confined within the structure, rather than the cover medium [31]. Therefore, the sensitivity and the FWHM are a pair of irreconcilable contradictions in sensors with passive gratings. Active gratings can easily solve the contradiction that passive gratings have. 

The coupling constant *κ* is defined as: (2)$$\kappa =\pi n_{real}/\lambda +\frac{1}{2}j\alpha ,$$ where *n*_real_ and *α* are the amplitudes of the spatial modulation and *λ* is the resonant wavelength of the DFB laser [6]. The gain modulation is represented by the imaginary part of the coupling constant. The above formula can be transformed into:(3)$$\frac{\lambda \kappa }{\pi }=n_{real}+\frac{\lambda }{2\pi }\alpha j.$$

Here, the term $\frac{\lambda \kappa }{\pi }$ can be regarded as the complex refractive index modulation, *n*. The term $\frac{\lambda }{2\pi }\alpha$ represents the gain modulation and can be regarded as the imaginary part of the refractive index modulation. In this case, the imaginary part of the refractive index (*n*_img_) represents the gain value of the active grating. The structural parameters are the same as in Section 2.1. We selected *d_g_* to be 200 nm (*d*_wg_ at 0 nm) and set DC at 40, 50 and 60%, which have maximum sensitivities of 218.2, 217.3 and 211.2 nm/RIU, respectively. The FWHM, relative intensity of the peak and the corresponding gain value (*n*_img_) of the gratings with DC = 40% and DC = 50% are depicted in Table 1 and Table 2, respectively.

The FWHMs are 0.13, 0.08 and 0.08 nm; the sensitivities are 230.0, 230.2 and 212.4 nm/RIU; the FOMs are 1769.2, 2877.5 and 2655.0 RIU^−1^ when the values of *n*_img_ are −0.1482, −0.1410 and −0.1107 and DCs are 40, 50 and 60%, respectively. If a narrower FWHM and a larger FOM are desired, these requirements can be implemented by choosing a more precise gain value.

As shown in Figure 3a, when the gain of the grating increases (the absolute value of the gain value |*n*_img_|), we can see that the relative intensity of the resonant wavelength grows slowly in the beginning, and after a critical point, starts to grow rapidly. After linear fitting the points in the two regions with different slopes, the fitting results are the same as the laser threshold curve. After calculating the abscissa of the intersection of the two fitted lines to be 0.11007, we can determine that the laser threshold gain is 0.11007. Figure 3b shows that the FWHM is proportional to |*n*_img_|. The larger |*n*_img_| is, the narrower the FWHM is. We chose five different values of *n*_img_ to reflect this trend of the resonant peak in Figure 3c. The electric field distribution for *n*_c_ = 1.33 is simulated and shown in Figure 3d. Comparing Figure 2c and Figure 3d, when the grating is set to *n*_img_ = −0.1107, the maximum electric intensity increases from 2.02 × 10^5^ to 9.69 × 10^7^ V/m, which indicates that more electric field is stored in the structure, leading to a narrower FWHM and a stronger relative intensity of the resonant wavelength. Using the obtained ultranarrow FWHM laser to perform the refractive index sensing simulation, the redshift of the resonant wavelength is shown in Figure 3e. The data in Figure 3f are from Figure 3e. Through linear fitting of the resonant wavelength to the refractive index, an FWHM of 0.080 nm, S of 212.4 nm/RIU, and a high FOM of 2655.0 RIU^−1^ are obtained when *n*_img_ = −0.1107 and DC = 60%. Therefore, without changing the structure, using an active grating can greatly reduce the FWHM and improve the FOM. The FWHM narrows from tens of nanometers to tens of picometers. These improvements can accurately simulate the performance of the sensors. The FOM of this design model is also more consistent with the FOM of the DFB laser sensor. To narrow the FWHM to a value of 0.1 nm, the sensors with DC = 40 and 50% require a value *n*_img_ of about −0.14, while a duty cycle of 60% requires only a value of −0.11. That is, the laser sensor with DC = 60% requires a lower gain with a similar sensitivity (230.0, 230.2 and 212.4 nm/RIU).

### 2.3. Sensors with Active Gratings for Biomolecule Detection

In the previous section, high FOM sensors were obtained by replacing passive gratings with active gratings. Here, we used active grating sensors to detect a biomolecule layer by monitoring the shift in wavelength. The parameters of the gratings were set to *d*_g_ = 200 nm (*d*_wg_ = 0 nm), DC = 40, 50 and 60% and *n*_img_ = −0.1482, −0.1410 and −0.1110. We assumed that the refractive index of the biomolecule layer (*n*_bl_) was 1.5. *d*_bl_ was set to 2, 4, 6, 8 and 10 nm. The other parameters are similar to those presented in Section 2.2. A schematic of the biosensor is depicted in Figure 4a, in which the outermost blue wrap layer represents the biomolecule layer. With the deposition of biomolecules on the surface and sides of the grating, the resonant wavelength of each sensor with different DCs is constantly redshifted, as shown in Figure 4b. Figure 4b shows that the wavelength redshift rate of sensors with different DCs is different. In the sensor with DC = 40%, depositing a 10 nm layer of biomolecules causes a redshift of the wavelength by 0.732 nm. In a 50% duty cycle, the redshift is 0.910 nm. The wavelength of the sensor under DC = 60% has a shift of 1.952 nm. The corresponding calculated sensitivities are: 0.07289, 0.09084 and 0.19499 nm/nm, respectively. The maximum wavelength shift is 1.22 nm more than the minimum one, and the sensitivity of the former is much larger than that of the latter. The increase in the wavelength shift or sensitivity is approximately 2.7 times. In Figure 2e, it can be seen that when all other parameters are kept constant, as the DC increases, the wavelength does not noticeably change when the DC is lower than 50%, but is quickly redshifted once the DC exceeds 50%. These findings explain why a DC = 60% sensor is 1.7 times more sensitive than a DC = 40% sensor. When biomolecules are deposited on both sides of the grating, this results in an increase in the DC of the grating. When the original duty cycle is less than 50%, the wavelength will not change significantly. If the previous duty cycle was higher than 50%, the increase in DC caused by biomolecule sedimentation would result in a large redshift. The wavelength redshift of a sensor with a large DC is much higher than the wavelength shift of a sensor with a small DC; that is, the sensitivity with a larger DC is higher. It can be seen that the active grating with DC = 60% is the better DFB laser sensor, which not only has the highest sensitivity but also the lowest laser threshold. Figure 4c shows the electric field distribution of the resonant wavelengths when a 10 nm layer of biomolecules is deposited. In the process of biomolecule deposition, the maximum electric intensity decreases from 1.03 × 10^8^ to 1.25 × 10^7^ V/m, which indicates that the deposition of the biomolecules causes some of the light energy to leak out of the structure, which is also consistent with the results seen in Figure 4d. In Figure 4d, it is shown that the deposition of biomolecules causes not only a shift in the wavelength, but also a decrease in the the relative intensity of the modes. Figure 4e is the two-dimensional normalized electric field distribution of the central axis of the structure (the red line in Figure 4c) with 0 and 10 nm layers of biomolecules, respectively. It can also be observed from Figure 4e that the point of the strongest electric field in the structure is moving from the grating towards the biomolecule layer, which will increase the diffraction loss and lead to the broadening of the FWHM. The shift of the electric field in the structure can be used to explain the results seen in Figure 4d.

## 3. Conclusions

In summary, the FWHM, sensitivity and FOM of passive grating sensors were theoretically analyzed. The results indicated that the FWHM and sensitivity are contradictory in the passive structure, meaning that the improvement of one must sacrifice the other. To avoid this problem, we replaced the passive grating with an active grating. The results showed that the FWHM decreased from 56.43 to 0.08 nm and the FOM increasd from 3.9 to 2844.5 RIU^−1^. Furthermore, we used the sensors with active gratings to detect biomolecules. After depositing a 10 nm layer of biomolecules, the sensor with DC = 60% redshifted 1.22 nm more than the sensor with DC = 40%, as the DC increase caused the biomolecule deposition on the grating sides. The maximum biological sensitivity was approximately 1.7 times more than the minimum. 

## Figures and Tables

**Figure 1 sensors-19-02569-f001:**
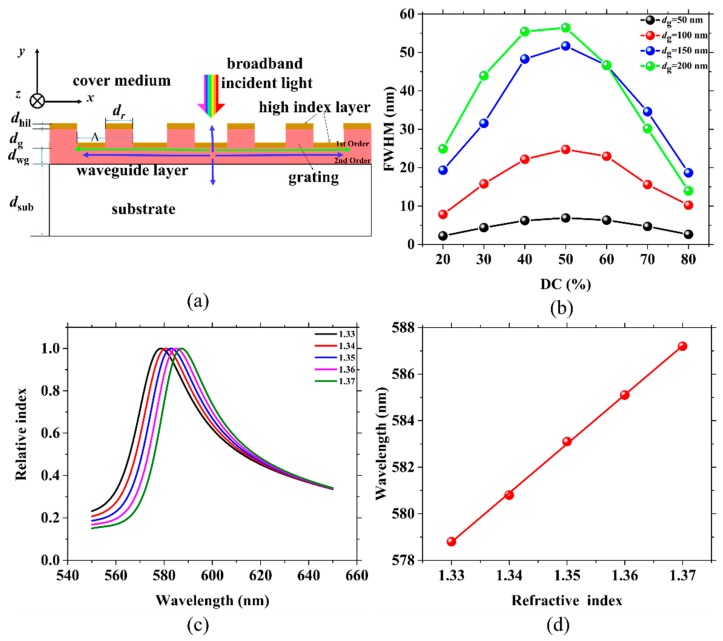
(**a**) Schematic structure of the passive grating sensors for wavelength shift detection. (**b**) The full width at half maximum (FWHM) for various groove depths (*d*_g_) and duty cycles (DCs); the same colored points represent identical values of *d*_g_; the *d*_g_ values are 50, 100, 150 and 200 nm. (**c**) An example of a resonant wavelength shift corresponding to a change in the index of refraction of the liquid medium (*n*_c_). (**d**) The linear fitting between the resonant wavelength (*λ*_peak_) and *n*_c_ in (c). DC = 60%, and the thicknesses of the waveguide layer (*d*_wg_) and *d*_g_ are 0 and 200 nm, respectively.

**Figure 2 sensors-19-02569-f002:**
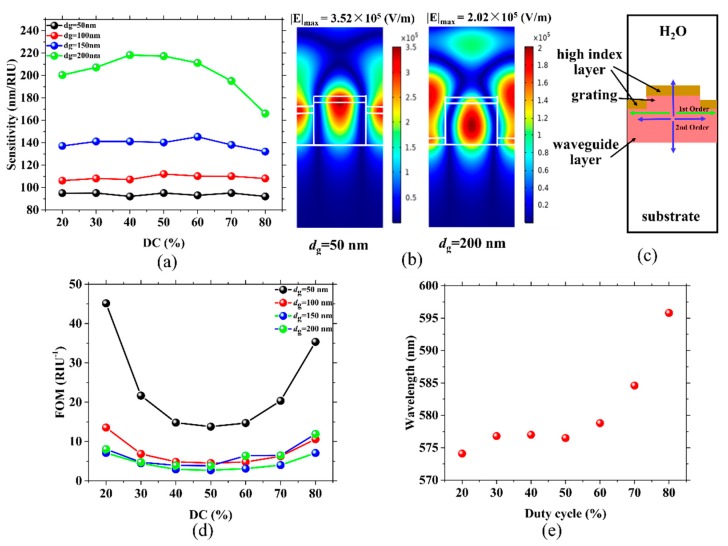
(**a**) Sensitivity versus *d*_g_ and DC; the same colored points represent the same values of *d*_g_. (**b**) Electric field distributions of the resonant wavelength, which are 627.5 and 577.8 nm for sensors with *d*_g_ = 50 nm (*d*_wg_ = 0 nm) and *d*_g_ = 200 nm (*d*_wg_ = 0 nm), respectively. In both cases *n*_c_ = 1.33, DC = 60% and the period of the grating (*Λ*) = 400 nm. (**c**) Schematic diagram of the waveguide structure. (**d**) The figure of merit (FOM) for various *d*_g_ and DCs; the same colored balls represent identical values of *d*_g_. (**e**) Calculated resonant wavelength versus duty cycle for a *d*_g_ of 200 nm.

**Figure 3 sensors-19-02569-f003:**
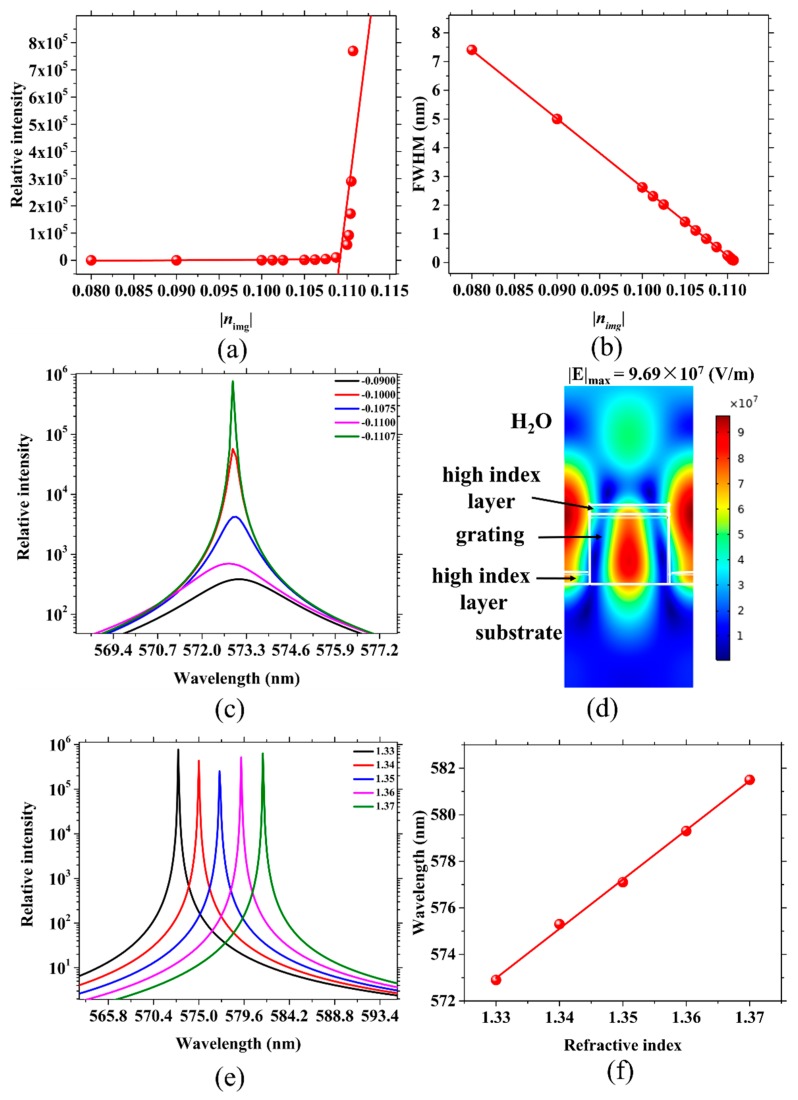
(**a**) The relationship between the relative intensity and the the absolute value of the active grating gain values (|*n*_img_|). (**b**) Corresponding FWHMs for various |*n*_img_|. (**c**) The variation trend of the resonant peak patterns reflected by five different values of *n*_img_. (**d**) Electric field distribution for a resonant wavelength of 572.9 nm with *n*_c_ = 1.33. (**e**) An example of the resonant wavelength shift corresponding to the change in *n*_c_. (**f**) The linear fitting between *λ*_peak_ and *n*_c_ in (**e**) with *d*_g_ = 200 nm, *d*_wg_ = 0 nm, DC = 60% and *Λ* = 400 nm.

**Figure 4 sensors-19-02569-f004:**
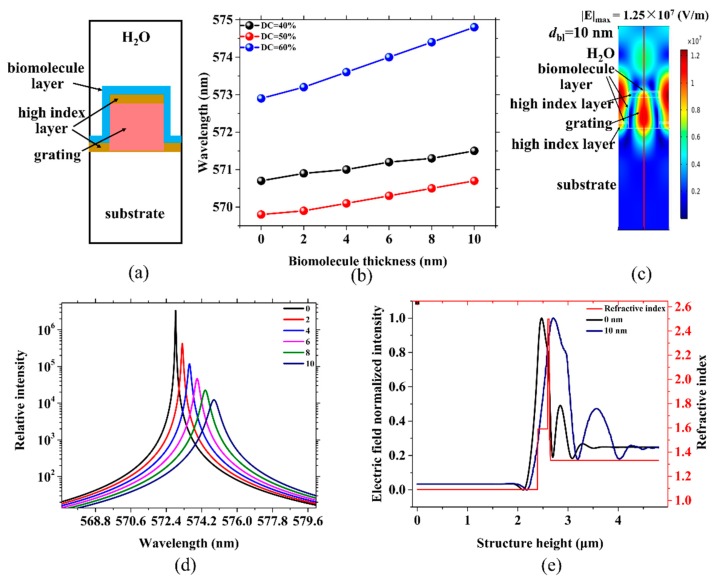
(**a**) Schematic of the sensor with the biomolecule layer. (**b**) The linear fitting between *λ*_peak_ and the thickness of the biomolecule layer (*d*_bl_) for DC = 40% (black), DC = 50% (red) and DC = 60% (blue). (**c**) Electric field distribution for a resonant wavelength of 574.9 nm and *d*_bl_ = 10 nm. (**d**) Resonant wavelength shift and relative intensity change corresponding to the deposition of biomolecules. (**e**) The refractive index distribution of the structure and two-dimensional normalized electric field distribution of the central axis of the structure (the red line in (**c**)) with 0 and 10 nm layers of biomolecules. The refractive indexes of the waveguide (*n_wg_*) and grating layers (*n_g_*) are 0 and 200 nm, respectively. The grating period and DC are 400 nm and 60%. The *n_c_* and biomolecule layer refractive indexes are 1.33 and 1.50, respectively.

**Table 1 sensors-19-02569-t001:** Relative intensity and FWHM for DC = 40% at different gain values (*n*_img_).

***n*_img_**	−0.1482	−0.1480	−0.1475	−0.1460	−0.1450	−0.1430
**Relative intensity**	306680	165010	61732	13141	7043	2968
**FWHM (nm)**	0.13	0.18	0.29	0.64	0.86	1.32
***n*_img_**	−0.1400	−0.1375	−0.1286	−0.1229	−0.1100	−0.100
**Relative intensity**	1251	746	216	123	48	28
**FWHM (nm)**	1.99	2.56	4.51	5.85	8.74	10.97

**Table 2 sensors-19-02569-t002:** Relative intensity and FWHM for DC = 50% at different gain values.

***n*_img_**	−0.1410	−0.1409	−0.1407	−0.1405	−0.1400	−0.1395
**Relative intensity**	770000	464040	226950	133970	52827	27984
**FWHM (nm)**	0.08	0.13	0.16	0.20	0.32	0.44
***n*_img_**	−0.1375	−0.1325	−0.1271	−0.1214	−0.1150	−0.1100
**Relative intensity**	6396	1437	423	203	109	73
**FWHM (nm)**	0.90	2.05	3.36	4.68	6.23	7.35

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
