# Peer review of "Design for Distributed Feedback Laser Biosensors Based on the Active Grating Model"

_sensors, 2019, doi:10.3390/s19112569_

Round 1
Reviewer 1 Report
This paper presents an idea of using active grating in a DFB laser biosensor to improve the sensitivity. The comparison between passive grating and active grating is made by the simulations of figure of merit (FOM)in both gratings. The simulation gives an important direction of performance improvement in such kind of biosensors. The work would be valuable for the relative research in this field though only simulation result is reported now.
However, the paper needs major revision before publication. The emphasis on the active grating and gain is right, but some results seems mainly based on the point around threshold point. It should be noted the threshold point is not a stable position for sensing because it normal results in large uncertainty around this point. The so-called intensity within the paper for such a point does not have too much value in this sense.
There are many points need to be improved. Some points are listed as follows.
1. Fig.2(b), the illustration of the electric field and the waveguide structure is not clear enough.
2. Captions of Fig.2 (c) and (d) are confusing.
3. In line 130, n real and α, λ0 and λ seems unclear. Please check.
4.It is not clear why the author used Table 1 and 2 other than figure like Figure 3.
5. The simulation results indicate that when DC is greater than 50%, the sensitivity increases sharply. The authors attribute this to the lower loss in the grating. Such an explanation is not convincing.
Reviewer 2 Report
-In line 55, it is written FMHM instead of FWHM
-In caption of Fig. 2, the Fig. 2c) is not described, and Fig.2d) is described twice where one of those corresponds to Fig.2c)
-In line 133, the term which describes the imaginary part of the refractive index modulation is not the same as the one of the equation 3. (The difference is the wavelength)
Reviewer 3 Report
The paper presents numerical investigations regarding DFB laser sensors, taking gain into account. One could argue, that without gain it cannot be called a DFB laser to begin with. The authors make the claim that "most of the theoretical design models are based on passive gratings", which is hard for me to verify. Overall I assume the article will be of low interest for most readers, but provides some methodology which can appeal to some.
I will present a list of modifications that I would suggest the authors to implement for a better article:
Line 30: "m=2 for vertically emitting from the surface": I do not see how the angle enters the discussion all of a sudden. No angles are mentioned in Eq. (1).
Line 38+: The authors list different methods how
other research groups have improved their DFB sensors, e.g. by
incorporating high index thin films. It is not entirely clear how this
improves the sensor, and the authors should expand this paragraph.
Line
45: Sensitivity has not been properly introduced. I assume it is d
lambda_peak/ dn for the first part, d lambda_peak/ d thickness for the
latter part. In any case the units of sensitivity and hence FOM are
inconsistent through the manuscript.
Line 55: FMHM: I assume this refers to FWHM and is just a misspelling. Otherwise this acronym needs to be introduced.
Line
56: The duty cycle is not explained. It is typically a/Lambda with a
being the length of either the dip of the plateau of the grating; or
which other structure is varied. This can be easily added to the figure
and be expanded.
Line 86: Why is "when" italic in the manuscript?
Figure 1(a): It might be a trivial thing, but the direction of propagation of the light is not specified, and should be added.
Figure
1(c): This is the first time the wavelength region is mentioned. No
explicit link in the text that ~570 nm is the target.
Figure
2(a): This plot is hard to read. One could argue that placing the DC at
the x-axis and use multiple colors for the 4 respective d_g data points
would be more visible. That is a matter of style though.
Figure 2(b): I am not familiar with the software used and hence have trouble understanding the color bars in the figure. I also advise indicating the direction of light propagation (despite maybe being trivial) and be consistent with the x-axis in Figure 1.
135+: The simulation uses
the imaginary part of the refractive index of the material to simulate
gain. Can this material actually provide gain? How realistic is the
assumption that such a material exists?
Table 1 and Table 2: Does the relative intensity refer to the peak of the relative intensity?
239: Why is the wavelength mentioned in bold font, and why has this wavelength been chosen?
Other remarks: the referencing seems inconsistent to me. Sometimes titles are placed in quotation marks, sometimes not. Looking at reference 1 the title immediately starts after the initials without separator.
Round 2
Reviewer 3 Report
The manuscript has been improved, unfortunately there are still some shortcomings in the manuscript. I will list them as before. The manuscript is already in a state it can be considered publishable, but improvements should improve readability.
Figure 1d: the y-axis corresponds to the peak wavelength of c, I assume. It would be better nomenclature to refer to lamba_peak instead of lambda, which has been defined as resonant wavelength/wavelength of the laser in text before.
Figure 1a, kind of: It is not clear from the Figure, how the sensor should work, as in where the light will be detected. In reflection vertical from the surface? From the waveguide? As only resonant modes are calculated this might not be important for the simulation, but if it should correspond to a physical device this would be interesting.
Similar remarks could be made about waveguiding effects in the other direction (z in the figure), but I assume that is ignored to reduce the dimensions in the simulation.
The color bars still have a strange labeling, such as 1.25x10^7x10^7 (Fig. 4c). The second factor probably refers to the scale of the bar and not the peak number, but the authors made it clear that the readability is sufficient for them. As the plot already provides value without colorbars I guess it is "good enough" as it is.
Equation 1 still does not contain an angle, whereas the consequences taken from it discuss such an angle. As an extra reference is given it is not impossible for the reader to follow up on this, especially since some help is given with the figure. Good enough.
Regarding response 8: It can be calculated from the equations that 560...600 nm is the interested wavelength range. Apparently the operating wavelength is not important enough to mention it explicitly, as per the author's decision.
Regarding response 11: I would have added such information to the manuscript to show the relevance of the simulation. It is not clear how far from reality the simulation is when effectively the gain is just chosen by changing the refractive index.
All in all the paper is publishable in its current form, but would benefit from additional polishing to attract more readers.
